# The Association between Osteoporosis and Grip Strength and Skeletal Muscle Mass in Community-Dwelling Older Women

**DOI:** 10.3390/ijerph16071228

**Published:** 2019-04-06

**Authors:** Yoshiaki Taniguchi, Hyuma Makizako, Ryoji Kiyama, Kazutoshi Tomioka, Yuki Nakai, Takuro Kubozono, Toshihiro Takenaka, Mitsuru Ohishi

**Affiliations:** 1Graduate School of Health Sciences, Kagoshima University, Kagoshima 890-8544, Japan; p.taniguchi0601@gmail.com (Y.T.); reha_tommy@yahoo.co.jp (K.T.); nakai@health.nop.kagoshima-u.ac.jp (Y.N.); 2Department of Physical Therapy, School of Health Sciences, Faculty of Medicine, Kagoshima University, Kagoshima 891-0133, Japan; 3Department of Physical Therapy, School of Health Sciences, Faculty of Medicine, Kagoshima University, Kagoshima 890-8544, Japan; kiyama@health.nop.kagoshima-u.ac.jp; 4Department of Rehabilitation, Tarumizu Municipal Medical Center, Tarumizu Chuo Hospita, Kagoshima 891-2124, Japan; 5Department of Cardiovascular Medicine and Hypertension, Graduate School of Medical and Dental Sciences, Kagoshima University, Kagoshima 890-0075, Japan; kubozono@cepp.ne.jp (T.K.); ohishi@m2.kufm.kagoshima-u.ac.jp (M.O.); 6Department of Internal Medicine, Tarumizu Municipal Medical Center, Tarumizu Chuo Hospital, Kagoshima 891-2124, Japan; takenaka@tarumizumh.jp

**Keywords:** osteoporosis, loss of skeletal muscle mass, older women

## Abstract

This cross-sectional study investigated the association between osteoporosis, grip strength, and skeletal muscle mass in community-dwelling older women. Data obtained from 265 older women who participated in a community-based health check survey (Tarumizu Study) were analyzed. Face-to-face interviews with participants revealed their history of osteoporosis. Appendicular skeletal muscle mass was assessed through bioelectrical impedance analysis, and appendicular skeletal muscle index was calculated. Dominant grip strength was also assessed. Loss of skeletal muscle mass (appendicular skeletal muscle mass < 5.7 kg/m^2^) and muscle weakness (grip strength < 18 kg) were determined based on criteria for sarcopenia put forth by the Asian Working Group for Sarcopenia. The prevalence rates of osteoporosis, muscle weakness, and loss of skeletal muscle mass were 27.2%, 28.7%, and 50.2%, respectively. Loss of skeletal muscle mass was more prevalent in participants with osteoporosis than in those without (65.3% vs. 44.6%, *p* < 0.01). The association between osteoporosis and muscle strength was not significant (30.6% vs. 28.0%, *p* = 0.68). After covariate adjustment, loss of skeletal muscle mass was found to be independently associated with osteoporosis (odds ratio 2.56, 95% confidence interval 1.33–4.91). In sum, osteoporosis was found to be associated with loss of skeletal muscle mass, but not with muscle weakness in community-dwelling older women.

## 1. Introduction

Osteoporosis is a worldwide health issue and is more common in women than men [1]. Loss of bone mass is a potential risk factor for fragility fractures, and osteoporotic fractures place a considerable burden on society [1].

The most recent National Livelihood Survey conducted by the Japanese Ministry of Health, Labour and Welfare indicated that falls and osteoporotic fractures were ranked fourth among the major causes requiring long-term care [2]. Among women, it is ranked third, accounting for 15.2% of the causes requiring long-term care [2]. The number of patients with osteoporosis in Japan was estimated to be 12.8 million in 2005 [3]. The prevalence rate of sarcopenia in Japan is 7.5–8.2% [4,5], and the prevalence rates of both sarcopenia and osteoporosis are expected to increase with the increase in the aged population. Sarcopenia increases the risk of falls, which can result in fragile fractures, including osteoporotic hip fractures [6,7]. Both sarcopenia and osteoporosis can lead to restrictions in daily living activities [8].

Muscle strength and muscle mass are the causes for mechanical loading, and they correlate to bone mineral density (BMD) [9]. Mechanical load by muscles is an essential mechanism for maintaining BMD [10]. Previous studies have reported that BMD is higher as the muscle mass increases [11,12,13]. In postmenopausal women, grip strength was related to BMD, independent of the relationship between appendicular skeletal muscle index (ASMI) and BMD [14]. Older women with strong muscle strength tended to have healthy bone status regardless of muscle mass [14].

It is well known that muscular strength and muscle mass influence BMD. Although several studies have shown the relationship between muscle mass and BMD, the relationship between the bone and the muscle of an individual already suffering from osteoporosis has not been clarified. The present study aims to investigate the relationship between muscle strength and muscle mass and osteoporosis (or its absence) among community-dwelling older women.

## 2. Materials and Methods

### 2.1. Participants

This current cross-sectional study used data from the Tarumizu Study, which was conducted in 2017. All variables were measured on the same day. The details of that study have been reported previously [15]. Briefly, it is a community-based health check survey conducted jointly with Kagoshima University (Faculty of Medicine), Tarumizu City Office, and Tarumi Chuo Hospital from November through December 2017. The present study excluded men (*n* = 97), participants who did not provide complete primary data (*n* = 17), and individuals aged less than 65 years (*n* = 1, measured before the 65th birthday). The final data for analysis were from 265 community-dwelling older women (mean age of 75.5 years) (Figure 1). This study was explained in advance to each participant by the approval of the Kagoshima University (Faculty of Medicine) Ethics Committee (Ref No. 170103), and informed consent was obtained.

### 2.2. Loss of Skeletal Muscle Mass and Muscle Weakness

To measure grip strength and muscle mass, we used the method proposed by Makizako et al. [15]. We assessed appendicular skeletal muscle mass (ASM) through a multi-frequency bioelectrical impedance analysis (BIA) using InBody 430 (InBody Japan, Tokyo, Japan). The InBody 430 analyzer adopts a tetrapolar, eight-point tactile electrode system that separately measures the impedance of the arms, trunk, and legs at three different frequencies (5, 50, and 250 kHz) for each segment. The surface of the hand electrode was placed in contact with each of the five fingers, while the participant’s heels and forefeet were placed on the circular foot electrode. The participants held out their arms and legs so that they would not contact any other body parts during the measurement. ASM was derived as the sum of the muscle masses of the four limbs, and the ASMI (kg/m^2^) was calculated. We categorized the participants into those with and without loss of muscle mass, to describe the prevalence of muscle mass loss for clinical interpretation. The grip strength of the dominant hand was assessed using a Smedley-type handheld dynamometer (GRIP-D; Takei Ltd., Niigata, Japan). The participants were divided into those with and without muscle weakness, similar to the categorization of loss of skeletal muscle mass [15]. Muscle weakness and loss of skeletal muscle mass was determined based on the criteria for sarcopenia derived by the Asian Working Group for Sarcopenia [16]; grip strength < 18 kg, ASMI < 5.7 kg/m^2^ for women.

### 2.3. Interviewer-Administered Questionnaire

Licensed doctors or nurses interviewed the participants about their osteoporosis (medical history or treatment), medications used, exercise habits (≥1 day/week), and fall history.

### 2.4. Walking Speed

Usual walking speed was used as the index of physical performance. We recorded the time taken to walk 10 m at usual walking speed and calculated usual walking speed. The walking speed was measured in seconds using a stopwatch. Participants were asked to walk on a flat and straight surface at a comfortable walking speed. Two markers were used to indicate the start and end of a 10-m walk path (measurement period), with a 2-m section to be traversed before passing the start marker (acceleration period), so that participants were walking at a comfortable pace by the time they reached the timed path. Participants were asked to walk another 2 m beyond the end of the path (deceleration period) to ensure a consistent walking pace on the timed path.

### 2.5. Statistical Analysis

Student’s *t*-tests and chi-square tests were used to test differences in characteristics, muscle mass, and grip strength between participants with osteoporosis and those without the disease. The prevalence rates of loss of skeletal muscle mass and muscle weakness in participants with osteoporosis were compared with those in participants without osteoporosis by using chi-square tests. In addition, loss of skeletal muscle mass associated with osteoporosis in the univariate analysis was examined as a dependent variable in the binomial logistic regression analysis. The adjusted model in the binomial logistic regression analysis included age, grip strength, usual walking speed, number of prescribed medications, exercise habits, and fall history as covariates. Adjusted odds ratios were estimated for osteoporosis, with 95% confidence intervals (CIs). All analyses were conducted using IBM SPSS Statistics 24.0 (IBM Corp., Armonk, NY, USA). The level of statistical significance was set at *p* < 0.05.

## 3. Results

### 3.1. Characteristics of the Study Participants

The characteristics of the participants and the results of comparisons of participants with osteoporotic participants and those without osteoporosis are shown in Table 1. Of the 265 participants, 72 (27.2%) exhibited osteoporosis. Participants with osteoporosis showed significantly lower weight (*p* = 0.015), lower body mass index (BMI) (*p* = 0.016), and greater use of medications used (*p* < 0.001) than those without osteoporosis.

### 3.2. Association between Osteoporosis and Muscle Mass and Muscle Strength

Participants with osteoporosis showed significantly lower ASMI compared with those without osteoporosis (*p* = 0.004). There was no significant difference in grip strength with or without osteoporosis (*p* = 0.154) (Table 1). Participants without osteoporosis compared to participants with osteoporosis had significantly decreased loss of skeletal muscle mass (65.3% vs. 44.6%, *p* = 0.004) (Figure 2). However, there was no significant difference in the prevalence of muscle weakness (30.6% vs. 28.0%, *p* = 0.680) (Figure 2).

The results of binomial logistic regression analyses are presented in Table 2. In the adjusted models, osteoporosis was independently associated with loss of skeletal muscle mass (odds ratio 2.56, 95% confidence interval 1.33–4.91).

## 4. Discussion

This cross-sectional study found a significant association between osteoporosis and loss of muscle mass. This association remained even after adjusting age, grip strength, usual walking speed, number of prescribed medicines, exercise habits, and fall history. However, no relationship was found between muscle weakness and osteoporosis among community-dwelling older women.

Osteoporosis and sarcopenia are two conditions associated with aging with similar risk factors, including genetics, endocrine function, and mechanical factors [11]. In the present study, we assessed muscle strength and muscle mass, which are parameters of sarcopenia, and examined their relationship with osteoporosis. In the results, only muscle mass was associated with osteoporosis. The relationship between muscle mass and BMD has been shown in previous studies. In a study targeting Japanese women, BMD and muscle mass of the lumbar spine and total hip were proven to be significantly related [17]. Other studies found that BMD and muscle mass bore relevance to each other [6,18,19,20].

Muscle mass is closely associated with bone mass. It has been suggested that the developed muscle may promote bone growth [21]. Also, muscle mass decline with aging appears to occur before bone mass decline with aging [18], suggesting that it may be possible to prevent osteoporosis by increasing muscle mass.

In the present study, there were differences among muscle mass, weight, and BMI, and the importance of the burden on bone by gravity was suggested for bone health. The relationships among weight, BMI, and BMD have also been found in previous studies. There is, however, controversy on whether higher fat mass positively or negatively affects fracture risk. A meta-analysis was published in 2005 including twelve prospective population-based cohort studies. Low BMI in both men and women increased any type of fracture and higher BMI decreased the risk of fracture [22]. In other studies, it was reported that being overweight and obese were significantly associated with a lower risk of all fractures in women [23]. Recent evidence suggests that higher fat mass may be associated with increased fracture risk, although it positively affects BMD [18]. Although body weight and BMI influence bone health as reported previously, the recommendation is to increase muscle mass rather than body fat mass.

Nutrition has been recognized as an important factor that can influence both muscle and bone [24]. We did not consider nutrition in the present study, however, significant differences in BMI due to osteoporosis or lack thereof have been found; therefore, nutritional factors may affect osteoporosis and muscle mass reduction.

The present study had several limitations. In this study, osteoporosis was investigated by self-reports. Therefore, it does not necessarily reflect BMD, which may also be low among participants who were not diagnosed with osteoporosis. Moreover, although previous studies suggest the impact of muscle mass on BMD, using a longitudinal design, the present cross-sectional study did not discuss the longitudinal association between osteoporosis and muscle weakness or loss of skeletal muscle mass. Furthermore, other potential covariates that can relate muscle strength and muscle mass, such as nutritional, life style, and hormonal factors, need to be considered. It is necessary to pay attention to the representativeness of the population sample as well. A total of 452 older people (representing approximately 10% of older people in the city) were enrolled in the study. The sample size was not calculated because this study is a city project, and all of the applicants participated. Moreover, they were not selected randomly. Future longitudinal studies are warranted.

## 5. Conclusions

The present study found a significant relationship between osteoporosis and loss of muscle mass. It may be possible to prevent osteoporosis and reduce the risk of fractures by increasing muscle mass.

## Figures and Tables

**Figure 1 ijerph-16-01228-f001:**
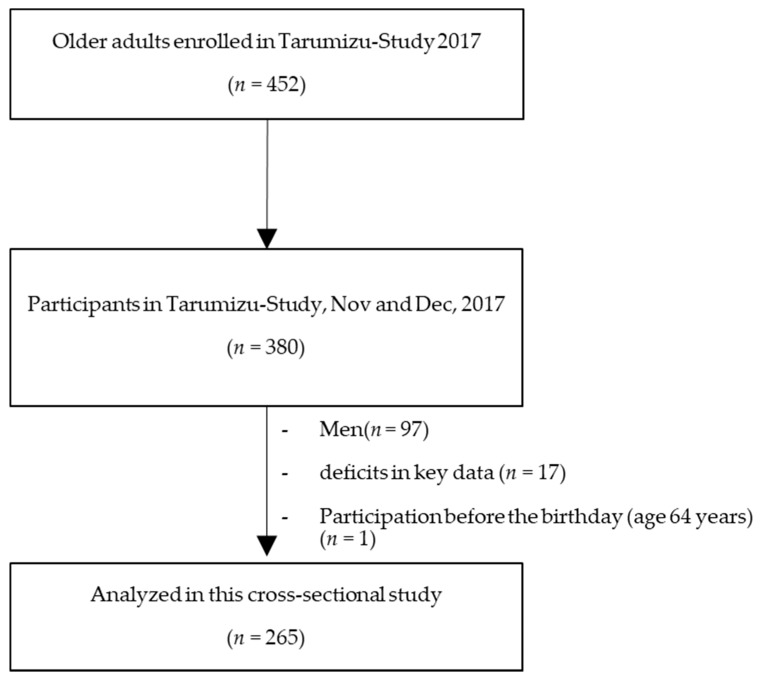
Participant inclusion criteria flow diagram.

**Figure 2 ijerph-16-01228-f002:**
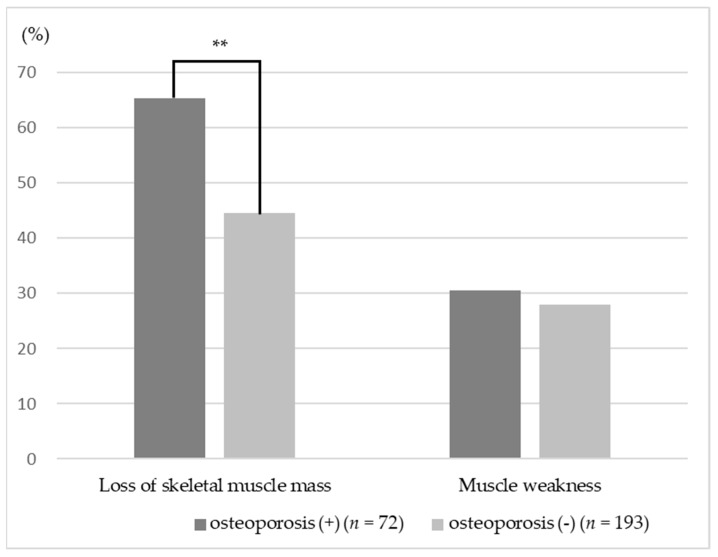
Ratio of loss of skeletal muscle mass and muscle weakness; ** *p* < 0.01.

**Table 1 ijerph-16-01228-t001:** Characteristics of the participants, mean ± SD, or %.

Characteristic	Over All (*N* = 265)	Participants with Osteoporosis (*N* = 72, 27.2%)	Participants without Osteoporosis (*N* = 193, 72.8%)	*p* ^a^
Age, years	75.5 ± 6.5	76.2 ± 6.4	75.3 ± 6.5	0.297
Height, cm	149.4 ± 5.5	149.1 ± 5.8	149.5 ± 5.4	0.593
Weight, kg	52.4 ± 8.8	50.3 ± 8.9	53.2 ± 8.6	0.015
Body mass index, kg/m^2^	23.5 ± 3.9	22.6 ± 3.6	23.8 ± 3.5	0.016
Medications, number/day	3.7 ± 3.9	5.6 ± 4.9	3.0 ± 3.2	<0.001
Grip strength, kg	20.4 ± 4.2	19.8 ± 4.1	20.6 ± 4.2	0.154
Usual walking speed, m/s	1.4 ± 0.2	1.4 ± 0.2	1.4 ± 0.4	0.285
ASMI, kg/m^2^	5.7 ± 0.7	5.5 ± 0.6	5.8 ± 0.7	0.004
No exercise habits, *n* (%)	42 (15.8%)	10 (13.9%)	32 (16.6%)	0.594
Fall history, *n* (%)	43 (16.2%)	9 (12.5%)	34 (17.6%)	0.315

SD, standard deviation; ASMI, appendicular skeletal muscle mass index, ^a^ Student’s *t*-test for continuous measures and *χ*^2^ test for proportions.

**Table 2 ijerph-16-01228-t002:** Odds ratios for loss of skeletal muscle mass associated with osteoporosis.

Variable	Dependent Value: Loss of Skeletal Muscle Mass	
	Odds Ratio (95% CI)	*p*
Osteoporosis	2.56 (1.33–4.91)	0.005
Age	1.03 (0.98–1.09)	0.257
Grip strength	0.79 (0.73–0.86)	<0.001
Usual walking speed	0.60 (0.14–2.57)	0.494
Medications	0.94 (0.86–1.03)	0.172
No exercise habits	1.46 (0.69–3.10)	0.327
Fall history	1.54 (0.72–3.32)	0.266

CI: Confidence interval.

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
