# Peer review of "The Association between Osteoporosis and Grip Strength and Skeletal Muscle Mass in Community-Dwelling Older Women"

_ijerph, 2019, doi:10.3390/ijerph16071228_

Round 1
Reviewer 1 Report
It’s fine to use a strength cut-point of<18kg, but different results may have been realized if a sample-specific value was used.
Information beyond the title and stated purpose is measured (ie, gait speed and sarcopenia). I can see its relevance but question the appropriateness of its inclusion.
It appears that information regarding osteoporosis was obtained by interview whereas information about strength and muscle mass were obtained by direct testing. That strikes me as problematic. Where, for example, and what procedure was used to extablish osteoporosis?
People-first language should be used (eg, “patients with osteoporosis” rather than “osteoporosis patients.”)
Passive voice should be eliminated (eg, “it was found”).
More information is needed regarding the measurement of skeletal muscle mass and strength.
I’m surprised by the results. Greater body weight and body weight index are usually accompanied by greater bone mineral density.
How was sample size determined?
The first column of Table 1 and Table 2 should be labeled. Much of the summary data in Table 1 is presented with excessive precision.
There are some problems with the references. Reference 13 only presents author initials. The article title for reference 20 is inappropriately capitalized.
Author Response
Manuscript ID: ijerph-463387
Type of manuscript: Article
Title: The Association between Osteoporosis and Grip Strength and Skeletal Muscle Mass in Community-Dwelling Older Women
Reviewer 1
Comment
It’s fine to use a strength cut-point of<18kg, but different results may have been realized if a sample-specific value was used.
Response
We appreciate your suggestion. As this study had a small sample size and only one cohort, we applied the Asian recommendations without using a cutoff specific for the cohort. We believe that it is reasonable to judge muscle weakness from grip strength because the lower one-third of the participants of this study had a grip strength of<18 kg.
Comment
Information beyond the title and stated purpose is measured (ie, gait speed and sarcopenia). I can see its relevance but question the appropriateness of its inclusion.
Response
We appreciate your comment. Walking speed was measured to confirm that there was no difference in physical performance. The information about sarcopenia was deleted.
Location in the text
p. 1, line. 28-29
The prevalence rates of osteoporosis, muscle weakness, and loss of skeletal muscle mass were 27.2%, 28.7%, and 50.2%, respectively.
p. 3, line. 105
The sentence has been deleted.
p. 3, line. 108-109
The prevalence rates of loss of skeletal muscle mass and muscle weakness in participants with osteoporosis were compared with those in participants without osteoporosis by using chi-square tests.
p. 4, line. 124
3.2. The association between osteoporosis and muscle mass and muscle strength
p. 4, line. 130
The sentence has been deleted.
p. 4, Table 1
The line on sarcopenia has been deleted
p.5, Figure 2.
The data on sarcopenia have been deleted from Figure 2.
p.5, line 133
Figure 2. Ratio of loss of skeletal muscle mass and muscle weakness; ** p < 0.01.
p. 5-6, line 144-146
In the present study, we assessed muscle strength and muscle mass, which are parameters of sarcopenia, and examined their relationship with osteoporosis. In the results, only muscle mass was associated with osteoporosis.
Comment
It appears that information regarding osteoporosis was obtained by interview whereas information about strength and muscle mass were obtained by direct testing. That strikes me as problematic. Where, for example, and what procedure was used to extablish osteoporosis?
Response
We appreciate your comment. In this study, we did not measure bone density. A doctor or a nurse asked each participant about whether she has a history of osteoporosis or was receiving treatment for osteoporosis.
Location in the text
p. 3, line. 94-95
Licensed doctors or nurses interviewed the participants about their osteoporosis (medical history or treatment), medications used, exercise habits (≥1 day/week), and fall history.
Comment
People-first language should be used (eg, “patients with osteoporosis” rather than “osteoporosis patients.”)
Response
We appreciate your suggestion. We rechecked the grammar and sentence structure including the use of people-first language, as you mentioned.
Location in the text
p. 2, line 45
The number of patients with osteoporosis in Japan was estimated to be 12.8 million in 2005 [3].
Comment
Passive voice should be eliminated (eg, “it was found”)
Response
We appreciate your suggestion. We rechecked the grammar and sentence structure, and eliminated the use of passive voice.
Location in the text
p. 2, line.54
In postmenopausal women, grip strength was related to BMD, independent of the relationship between appendicular skeletal muscle index (ASMI) and BMD [14].
Comment
More information is needed regarding the measurement of skeletal muscle mass and strength.
Response
We appreciate your suggestion. We have added sentences to provide more information about the measurements of skeletal muscle mass and grip strength.
Location in the text
p. 3, line. 78-90
We assessed appendicular skeletal muscle mass (ASM) through a multi-frequency bioelectrical impedance analysis (BIA) using InBody 430 (InBody Japan, Tokyo, Japan). The InBody 430 analyzer adopts a tetrapolar, eight-point tactile electrode system that separately measures the impedance of the arms, trunk, and legs at three different frequencies (5, 50, and 250 kHz) for each segment. The surface of the hand electrode was placed in contact with each of the five fingers, while the participant’s heels and forefeet were placed on the circular foot electrode. The participants held out their arms and legs so that they would not contact any other body parts during the measurement. ASM was derived as the sum of the muscle masses of the four limbs, and the ASMI (kg/m2) was calculated. We categorized the participants into those with and without loss of muscle mass, to describe the prevalence of muscle mass loss for clinical interpretation. The grip strength of the dominant hand was assessed using a Smedley-type handheld dynamometer (GRIP-D; Takei Ltd, Niigata, Japan). The participants were divided into those with and without muscle weakness, similar to the categorization of loss of skeletal muscle mass.[15]
Comment
I’m surprised by the results. Greater body weight and body weight index are usually accompanied by greater bone mineral density.
Response
We appreciate your comment. We rechecked the grammar and sentence structure including the point you mentioned.
Location in the text
p. 3-4, line. 121-122
Participants with osteoporosis showed significantly lower weight (p = 0.015), lower body mass index (BMI) (p = 0.016), and greater if medications used (p < 0.001) than those without osteoporosis.
Comment
How was sample size determined?
Response
We appreciate your question. The sample size was not calculated because this study is not an interventional study. The participants of this study were selected from about 3810 older people living in Tarumi, a local city in Kagoshima, Japan. They were recruited through local newspaper advertisements and community campaigns. A total of 452 older people were enrolled, and 380 of them participated in this study.
Comment
The first column of Table 1 and Table 2 should be labeled. Much of the summary data in Table 1 is presented with excessive precision.
Response
Thank you for the detailed check. We labeled the first column of Tables 1 and 2. The values in Table 1 were rounded to the first decimal place.
Location in the text
Characteristic | Over all | Participants with | Participants without osteoporosis | pa | |||
Age, years | 75.5±6.5 | 76.2±6.4 | 75.3±6.5 | 0.297 | |||
Height, cm | 149.4±5.5 | 149.1±5.8 | 149.5±5.4 | 0.593 | |||
Weight, kg | 52.4±8.8 | 50.3±8.9 | 53.2±8.6 | 0.015 | |||
Body mass index, kg/m2 | 23.5±3.9 | 22.6±3.6 | 23.8±3.5 | 0.016 | |||
Medications, number/day | 3.7±3.9 | 5.6±4.9 | 3.0±3.2 | <0.001< span=""> | |||
Grip strength, kg | 20.4±4.2 | 19.8±4.1 | 20.6±4.2 | 0.154 | |||
Usual walking speed, m/s | 1.4±0.2 | 1.4±0.2 | 1.4±0.4 | 0.285 | |||
ASMI, kg/m2 | 5.7±0.7 | 5.5±0.6 | 5.8±0.7 | 0.004 | |||
No exercise habits, n (%) | 42(15.8%) | 10(13.9%) | 32(16.6%) | 0.594 | |||
Fall history, n (%) | 43(16.2%) | 9(12.5%) | 34(17.6%) | 0.315 | |||
SD, standard deviation; ASMI, appendicular skeletal muscle mass index. | |||||||
a Student’s t-test for continuous measures and χ2 test for proportions | |||||||
Dependent value: | |||||||
Variable | Odds ratio (95% CI) | p | |||||
Osteoporosis | 2.56 (1.33-4.91) | 0.005 | |||||
Age | 1.03 (0.98-1.09) | 0.257 | |||||
Grip strength | 0.79 (0.73-0.86) | <0.001< span=""> | |||||
Usual walking speed | 0.60 (0.14-2.57) | 0.494 | |||||
Medications | 0.94 (0.86-1.03) | 0.172 | |||||
No exercise habits | 1.46 (0.69-3.10) | 0.327 | |||||
Fall history | 1.54 (0.72-3.32) | 0.266 | |||||
CI, confidence interval | |||||||
Comment
There are some problems with the references. Reference 13 only presents author initials. The article title for reference 20 is inappropriately capitalized.
Response
Thank you for the detailed check. We have corrected reference 13. The capitalization of the article title in reference 20 has been corrected (initial capital letter only).
Location in the text
p. 7, line. 223
Pluijm, S.M.; Visser, M.; Smit, J.H.; Popp-Snifders, C.; Roos, J.C.; Lips, P. Determinants of bone mineral density in older men and women: Body composition as mediator. J. Bone Miner. Res. 2001, 16, 2142–2151.
p. 7, line. 240-241
Kim, S.; Won, C.W.; Kim, B.S.; Choi, H.R. The association between the low muscle mass and osteoporosis in elderly Korean people. J. Korean Med. Sci. 2014, 995–1000.26.

Reviewer 2 Report
The paper presented by Taniguchi Yoshiaki and colleagues reports data about a cross-sectional study, aiming to find a possible correlation between osteoporosis and sarcopenia. The Authors found an association between osteoporosis and skeletal muscle mass loss, but not with strength.
Even if the paper is simple, it is carefully written, the objective is clear, and the results are sufficiently well exposed. Moreover, the Authors correctly report several limitations due to the study design, which pose some cautions in generalizing the reported results.
However, there are some issues to be addressed:
1) Materials and methods (L 82-83) and Discussion (L 160-163): osteoporosis assessment was based on self-reports. What does this mean exactly? That no further analysis was made, or was the judgement based on previous diagnosis?
2) Materials and methods (L 84-92): when was walking speed measurement performed? From the cited paper (ref. 15), it is possible to get that grip-strength was already measured, but there’s no mention about walking speed. Was it assessed after the enrollment of the subjects for this study, or was is performed back in 2017?
3) Materials and methods (L 97-98): was a normality test performed before using the Student’s t-test? Student’s t-test requires that data are normally distributed. Moreover, since different characteristics were considered (age, height, weight, BMI, medications, grip strength, walking speed, ASMI), was multiple comparison correction applied? If not, why?
4) Results (L 111-113): based on the data reported in table 1, the sentence is inaccurate. It should be like “Participants with osteoporosis showed significantly lower weight (p=0.015), lower body mass index (BMI) (p=0.016), and greater if medications used (p<0.001) compared with those without osteoporosis”
Author Response
Manuscript ID: ijerph-463387
Type of manuscript: Article
Title: The Association between Osteoporosis and Grip Strength and Skeletal Muscle Mass in Community-Dwelling Older Women
Reviewer 2
Comment
1) Materials and methods (L 82-83) and Discussion (L 160-163): osteoporosis assessment was based on self-reports. What does this mean exactly? That no further analysis was made, or was the judgement based on previous diagnosis?
Response
We appreciate your comment and questions. The participants were judged to have osteoporosis if they reported having osteoporosis or being treated for the condition in their interview with the doctor or nurse.
Location in the text
p. 3, line. 94-95
Licensed doctors or nurses interviewed the participants about their osteoporosis (medical history or treatment), medications used, exercise habits (≥1 day/week), and fall history.
Comment
2) Materials and methods (L 84-92): when was walking speed measurement performed? From the cited paper (ref. 15), it is possible to get that grip-strength was already measured, but there’s no mention about walking speed. Was it assessed after the enrollment of the subjects for this study, or was is performed back in 2017?
Response
We appreciate your suggestion. We added a sentence about walking speed measurement.
Location in the text
p. 2, line. 65
All variables were measured on the same day.
Comment
3) Materials and methods (L 97-98): was a normality test performed before using the Student’s t-test? Student’s t-test requires that data are normally distributed. Moreover, since different characteristics were considered (age, height, weight, BMI, medications, grip strength, walking speed, ASMI), was multiple comparison correction applied? If not, why?
Response
We appreciate your questions. We confirmed that the main outcomes (grip strength and ASMI) were normally distributed. In this study, binomial logistic regression analysis was mainly conducted, and differences between groups were confirmed only in univariate analysis.
Comment
4) Results (L 111-113): based on the data reported in table 1, the sentence is inaccurate. It should be like “Participants with osteoporosis showed significantly lower weight (p=0.015), lower body mass index (BMI) (p=0.016), and greater if medications used (p<0.001) compared with those without osteoporosis”
Response
We thank you for your point. We have changed the sentence according to your suggestion.
Location in the text
p. 3, line. 121-122
Participants with osteoporosis showed significantly lower weight (p = 0.015), lower body mass index (BMI) (p = 0.016), and greater if medications used (p < 0.001) than those without osteoporosis.

Round 2
Reviewer 1 Report
I am still bothered by the use of self-report as an indicator of osteoporosis.
Sample size calculation is not just for interventional studies.
On page 4 (line 122) “greater” what?
On page 5 (line 151) “Muscle growth grows faster than bone” makes no sense.
Passive voice (particularly “It” statements persist.
Author Response
Manuscript ID: ijerph-463387
Type of manuscript: Article
Title: The Association between Osteoporosis and Grip Strength and Skeletal Muscle Mass in Community-Dwelling Older Women
Reviewer 1
Comment
I am still bothered by the use of self-report as an indicator of osteoporosis.
Response
We appreciate your suggestion. According to previous research, the prevalence of osteoporosis prescription and self-reported osteoporosis was moderate to good [1]. Therefore, I think that the diagnoses and self-reports of osteoporosis do not change greatly. However, BMD may be low, even in people who have not been diagnosed with osteoporosis, as described in the limitations.
1. Peeters, G. M.; Tett, S. E.; Dobson, A. J.; Mishra, G. D., Validity of self-reported osteoporosis in mid-age and older women. Osteoporos. Int. 2013, 24, (3), 917-27.
Location in the text
p. 6, line. 170-172
In this study, osteoporosis was investigated by self-reports. Therefore, it does not necessarily reflect BMD, which may also be low among participants who were not diagnosed with osteoporosis.
Comment
Sample size calculation is not just for interventional studies.
Response
We appreciate your suggestion. This study was a project of the city, and all of the applicants were analyzed. Therefore, the sample size was not calculated, and the target number of participants were judiciously decided upon. We added this information in the limitations section.
Location in the text
p. 6, line. 178-180
The sample size was not calculated because this study is a city project, and all of the applicants participated.
Comment
On page 4 (line 122) “greater” what?
Response
We appreciate your suggestion. "Greater" refers to the use of many medications. I have changed the phrase “greater if medications used” to “greater use of medications.”
Location in the text
p. 4, line. 122
Participants with osteoporosis showed significantly lower weight (p = 0.015), lower body mass index (BMI) (p = 0.016), and greater use of medications (p < 0.001) than those without osteoporosis.
Comment
On page 5 (line 151) “Muscle growth grows faster than bone” makes no sense.
Response
We appreciate your comment. "Muscle growth grows faster than bone" has been deleted and the text has been changed.
Location in the text
p. 5, line. 151-152
It has been suggested that the developed muscle may promote bone growth [22].
Comment
Passive voice particularly “It” statements persist.
Response
We appreciate your suggestion. I changed the text in lines 170-172.
Location in the text
p. 6, line. 170-172
In this study, osteoporosis was investigated by self-reports. Therefore, it does not necessarily reflect BMD, which may also be low among participants who were not diagnosed with osteoporosis.
